# Management of acute acquired comitant esotropia using prisms and vision therapy

PremNandhini Satgunam[1,2]*, Rinki Gupta[1,3], Manali Sikder[1,3], Rohan Nalawade[3], Ramesh Kekunnaya[3]

1 Brien Holden Institute of Optometry and Vision Sciences, Hyderabad, India, 2 Hyderabad Eye Research Foundation, L V Prasad Eye Institute, Hyderabad, India, 3 Child Sight Institute, Jasti V Ramanamma Children's Eye Care Center, L V Prasad Eye Institute, Hyderabad, India

☺ These authors contributed equally to this work.

* premnandhini@lvpei.org

## Abstract

### Purpose

Acute acquired comitant esotropia (AACE) once considered rare, is now becoming more prevalent. Surgery in these cases can get delayed when the strabismus angle varies or if the patient is unwilling. In such cases optical management and vision therapy can be considered. We present the outcomes of such management for AACE.

### Methods

This retrospective study included a cohort of patients diagnosed with non-accommodative or non-neurological AACE, managed in our institute between January to October, 2023. Prisms were prescribed to patients experiencing constant double vision. All patients were given vision therapy exercises. Eso deviation and divergence range were measured pre and post therapy.

### Results

Fourteen patients were included. Of these, 5 patients (with eso deviation ≤20PD) were relieved with vision therapy exercises alone. For the remaining 9 patients, prisms were prescribed along with vision therapy. Post-vision therapy, 2 patients could achieve binocular single vision without prisms, and in 4 patients the prism power was reduced. In the remaining 3, full prism correction was warranted, one of them opted for surgery. The median reduction in eso deviation by 6.5PD (range:2–25PD) and increase in divergence by 7PD (range:0–17PD) after vision therapy was significant (p < 0.03).

### Conclusion

Non-surgical intervention reduced the eso deviation in 79% (11/14) of patients. Certain types of AACE seem to be amenable to vision therapy and/or optical

**Data availability statement:** All the relevant data are contained within the manuscript.

**Funding:** The study was funded by the Hyderabad Eye Research Foundation. This funding was received by Dr. PremNandhini Satgunam.

**Competing interests:** The authors have declared that no competing interests exist.

management. Temporary Fresnel prisms can be considered in majority of these patients and tapered away later, if their divergence range can be improved with vision therapy.

## Introduction

Acute acquired comitant esotropia (AACE) presents with double vision usually of sudden onset, with inward deviation of the eye [1,2]. The inward deviation can alternate between either eye. The deviation can be constant or intermittent. The condition can occur in both children and adults [3,4]. AACE was previously reported to be relatively rare [5], however in recent years, its incidence has increased [6]. Excessive near work, particularly with digital devices is considered as a potential etiological factor for this increase [6–8]. Typically, the management of AACE includes Botulinum toxin injection and surgery [2,9]. Prisms are also considered for relieving double vision [10]. Few studies have shown prisms can also be tapered gradually [1,11]. Similar to the reported trends, we have also observed an increase in cases of AACE in our institute. We found that the number of new cases of AACE reported in 2018 at our referral tertiary eye care institute was about 1 in 1000 and in 5 years this increased to 2 in 1000.

The decision for surgical alignment of the eyes is usually taken by the strabismologist after assessing the deviation on different visits within a time period of up to 6 months. During these visits, the variability (or lack thereof) in the eye deviation is determined. In the interim, or if the patient is not ready for surgery, they are managed by patching to avoid diplopia or given prisms to aid fusion or botulinum toxin injection to reduce the eso deviation. For prism trial and prescription, patients are cross-referred to the Binocular Vision & Orthoptics clinic within the institute. In this clinic, in addition to prescribing prisms, patients are also given vision therapy exercises (in-office or home based) to improve their divergence range once single binocular vision (with or without prisms) has been achieved. Given that these patients previously demonstrated binocular single vision, training with vision therapy to revert to the prior condition, renders vision therapy plausible. It has been observed that eso deviation acquired in spasm of near reflex, can be relieved with optical measures and vision therapy [12] In a similar vein, in this study we present the outcomes of a non-surgical intervention for AACE patients. The management protocol of using prisms and vision therapy, to relieve diplopia and decrease the eso deviation are also described.

## Materials and methods

A retrospective study was conducted. The study protocol adhered to the tenets of Declaration of Helsinki and was approved by the institutional review board. Medical records of patients who visited the Binocular Vision & Orthoptics clinic from 01/01/2023 to 31/10/2023 were reviewed. Clinical photographs of patients were taken with informed consent. Data collected was anonymized. Patients referred to this clinic with a diagnosis of AACE (made by the pediatric and/or neuro-ophthalmologists) were included. Amongst these patients, the following types of AACE, as per the

newer classification [2,13], types I (Swan, occlusion-related) and II (Burian-Franceschetti, Bielschowsky, idiopathic) were included and types III (accommodative), IV (decompensated, post-surgical), V (neurologic), VI (cyclic) and VII (secondary) were excluded. Although this classification was originally developed for children, we applied this classification in our study that included both adults and children. Patients with other comorbidities were excluded. Patient's history was documented, including smartphone use particularly in supine position. This question was asked, since we have observed patients to give such a history of using smartphones in supine position for long durations, especially during the COVID-lockdown time and presenting with eso deviation [7]. The methods described herein are the typical clinical tests conducted in this clinic.

## Binocular vision assessment

All patients underwent cycloplegic refraction (one drop of 1% cyclopentolate hydrochloride and one drop of tropicamide plus (0.8% tropicamide +5% phenylephrine hydrochloride)) in their initial visit.. Binocular vision assessment was performed on a separate visit, with the best refractive correction. The measurements included: stereoacuity for near with Randot stereo test (Stereo Optical Company, Inc.), qualitative measures of fusion with the Worth-four dot test at 40 cm and a red lens test over one eye to examine distance fusion. The red lens test has lower dissociation when compared to the Worth-four dot test [14]. In this test, the red lens is placed over one eye, and the patient is asked to look at the distance acuity chart (computerized visual acuity chart) and report the number of charts seen and the color of the chart, under normal viewing condition with room lights on. In the presence of diplopia, a white and a red chart would be seen distinctly separately. If fusion is present, the patient would report seeing one chart with a light red tinge. Alternate prism cover test was performed for distance and near. If the eso deviation for distance and near was within 4PD (prism diopter), it was termed a basic eso deviation. If the eso deviation was greater than 4PD for distance than near, it was termed divergence insufficiency. If the eso deviation was greater for near it was termed convergence excess [15].

Subjective measures of eye deviation were also made with the Vision Therapy System version 4 (VTS 4, HTS inc., Gold Canyon, AZ, USA). The VTS4 has an integrated 3D screen with stereo goggles that allows dichoptic presentation of the stimuli [16]. The "phoria" module in VTS4 was used, where the patient was asked to align the plus to the immovable circle, using buttons on an X-box (Fig 1a). The testing distance was set at 84 cm, and the patient wore stereo-goggles. The angle of deviation was the distance between the plus and the circle, and this was displayed in prism diopters. While the prism cover test gives the objective measure, with VTS4 the subjective angle of deviation can be measured. With VTS4, magnitudes beyond that measured with the Muscle Imbalance Measure/Thorington card, and simultaneous measure of both horizontal and vertical deviation can be obtained.

Vergence ranges (convergence & divergence) were measured with a prism bar (distance and near). Vergence ranges were also measured using the VTS4 system, if the patient could maintain fusion at the testing distance. In this test, dichoptic targets of a dolphin with lines around it on 4 sides (up, down, left and right) were shown. The patient was asked to press the side of the "odd one". While on 3 sides the lines would appear as a plus, only on one side it will not appear as a plus (odd one) (Fig 1b). If the odd one was detected correctly, the vergence demand was increased automatically till the patient reported seeing two dolphins, at which point, the judgment of the odd one was difficult. This point is taken as the break point. The vergence demand was then automatically decreased, until the patient regained fusion to detect the odd one. This point was taken as the recovery point. Besides vergence, accommodation was also measured to ascertain accuracy with dynamic retinoscopy (monocular estimation method, MEM). Other measures of accommodation such as amplitude of accommodation and accommodative facility were also documented.

## Prism correction

Fresnel prisms were prescribed to alleviate constant diplopia. The smallest prism diopter that aided fusion for distance and near was prescribed. Typically, the Fresnel prism is dispensed only over one eye as it hampers contrast and reduces visual acuity. [17] However, we modified this protocol. The prism was split equally between the two eyes, so that the same

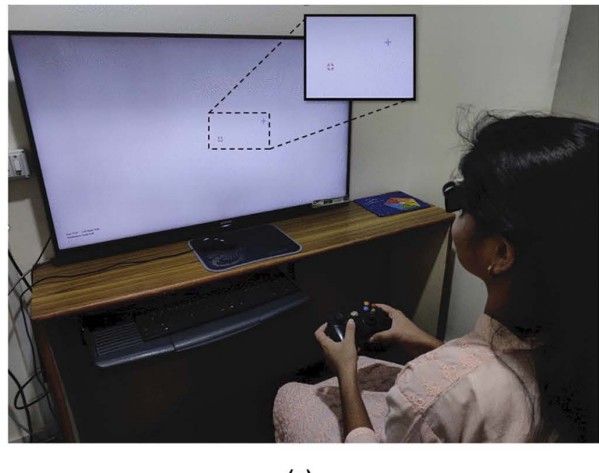
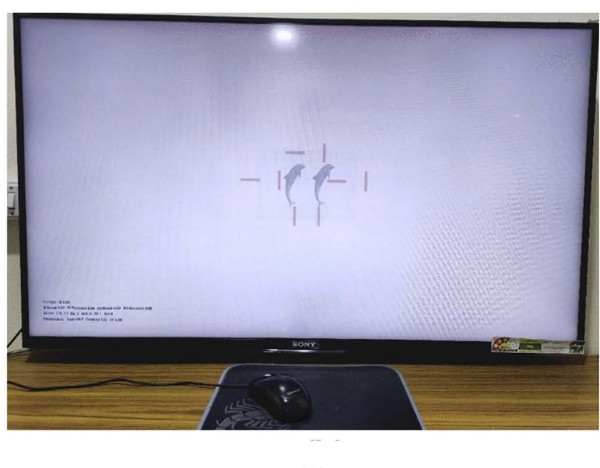

**(a)**                                                  **(b)**

**Fig 1. VTS4 system includes a stereo-television with dichoptic targets viewed through stereo-goggles. (a)** phoria measurement requiring alignment of the blue cross into the red circle using an X-box. The inset shows the magnified view of the dichoptic targets. **(b)** Flat fusional range measurement task. Through the dichoptic stereo-goggles, only one dolphin would be seen when fused, with lines around it. The patient was asked to identify the odd side. In this image, the odd target is below. In the remaining three sides, a red plus is seen with fusion.

Fresnel sheet was cut into two and stuck on each spectacle lens. For example, if 30PD base out eliminated the diplopia, then one 15PD Fresnel sheet was cut and dispensed such that each eye received 15PD base out. Splitting the Fresnel prism between the two eyes was economical for the patient and easy to dispense. Patients were given vision therapy for divergence exercises while wearing the prisms. When the patient improved in their divergence amplitude with vision therapy, it was possible to peel off one of the Fresnel sheets, to reduce the prism correction by half. With the reduced prism correction, the patient continued the divergence exercises. During subsequent visits, complete removal or reduction of prism magnitude were considered..

## Vision therapy protocol

The battery of divergence exercises for in-office therapy included double aperture rule, vectogram, VTS4 divergence exercises at different viewing distances (84 cm, 2m, 3m), accommodation exercises (included due to the coupling between accommodation and vergence systems), and Brock string. The typical in-office vision therapy was given for a duration of 45–60 minutes in a session (see also Vision Therapy Protocol document in supporting information). Patients were also encouraged to do the Brock string exercise at home (for 10–20 minutes per day). For those who could not come for in-office therapy only the Brock string exercise was given for a minimum duration of 20 minutes every day, twice a day. These patients were given an orientation to this exercise during their in-person clinic visit.

With the Brock string, the patients were trained to appreciate physiological diplopia. They were asked to maintain the fixated bead single for 20 seconds and encouraged to move that bead farther away until it was perceived as double. At this break point, they were encouraged to attempt to fuse the bead into a single bead. When not possible, the bead was brought in a little closer to achieve single vision, and again from that distance, the patient was encouraged to take the bead back, maintaining clear, single vision. Patients were instructed to monitor the break point distance and to increase that distance every day. Vergence rock exercises were also combined with the Brock string. Patients were asked to look at the beads at three different distances, while maintaining clear and single vision at each fixated bead for about 20s, while appreciating physiological diplopia. Such an exercise would improve the vergence facility to make eye movements without noticing the double vision and locking fusion seamlessly at different viewing distances.

For in-office therapy, usually 10 sittings that can be completed in 2-weeks or spread over few weeks (depending on patient's ability to come) were given. Binocular vision parameters were measured before (baseline) and after the vision therapy sessions.

## Data analysis

The primary outcome parameters to assess improvement were eso deviation measured before and after therapy (with and without prism spectacles), magnitude of negative fusional vergence (or divergence amplitude) before and after therapy. All measurements were made for both distance (3m) and near (40 cm). A non-parametric test for paired comparisons (Wilcoxon Signed Ranks test) was performed in SPSS version 21 (IBM Corporation) due to the non-normality of the data (Kolmogorov-Smirnov test, $p < 0.05$).

## Results

Thirty-four patients with diagnosis of AACE visited the Binocular Vision and Orthoptics clinic during the study period of 10 months. Table 1 displays the demographic details of these patients. There were significantly ($x^2 = 4.2$, df = 1, p = 0.04) more males (n = 23, 68%). Of the 34, 14 patients completed vision therapy (in-office and/or home vision therapy) exercises and came for their follow up visit. Others did not meet the inclusion criteria, had or opted for surgery, did not undergo vision therapy or were lost to follow-up. Only the results of the 14 patients (64% males) who completed the therapy are discussed further (Table 2). All of these patients had Type II AACE. Lag of accommodation was documented in 12 patients, and no accommodative spasm was observed in any of the patients.

Of the 14 patients, five (36%) did not require any prism correction. In these patients the distance eso deviation ranged from 12 to 20PD, and they only had intermittent, not constant diplopia. In the remaining nine patients, for eight, temporary Fresnel prism was dispensed to manage their constant diplopia, and one patient preferred ground prisms for cosmetic reasons. The eso deviation for distance in these patients was ≥ 20PD. After vision therapy, in six patients the prescribed Fresnel prism magnitude could be reduced (n = 4) or removed (n = 2). No patient complained of diplopia after vision therapy and or with prism corrections. The median reduction in eso deviation for all patients at the end of therapy (without the prism correction if any) was 6.5PD (range: 2–25PD) for distance and 6PD (range: 0–29PD) for near. Both of these reductions were significant (Wilcoxon signed rank, $p < 0.009$). Fig 2 shows the distribution of the eso deviation measured with cover test for both distance and near, with and without the prism prescription (if any). While performing the cover test with prism prescription, the neutralizing prism was held over the eye without the Fresnel prism. We also observed that holding the prism in the eye with Fresnel prescription gave similar values.

**Table 1. Demographics of all the patients (n = 34), along with the sub-group of patients (n = 14) who were able to complete the vision therapy.**

|  | 34 patients | 14 patients |
|---|---|---|
| Mean Age± standard deviation (SD) (years) | 24 ± 9.9, Range: 8–53 | 22 ± 8.8, Range 10–37 |
| Males: Females (n) | 23:11 | 9:5 |
| Digital Device (DD) use (hours/day), Mean ± SD | 8 ± 3.96, Range: 1–14 | 7 ± 3.93, Range: 1–14 |
| DD use in supine position yes: no: not available (n) | 19:3:12 | 7:2:5 |
| Duration of onset of the condition (months) | Range: 2–96 | Range: 4–48 |
| Distance eso deviation (prism diopters), Mean ± SD | 27 ± 14, Range: 8–55 | 28 ± 12, Range: 12–45 |
| Basic eso deviation (n) | 23 | 9 |
| Divergence Insufficiency (n) | 8 | 3 |
| Convergence excess (n) | 3 | 2 |

**Table 2. Clinical profile of the 14 patients included in the study, before and after therapy (prism and/or vision therapy).**

| S. No | Age (years)/ Gender | SE (DS) RE/LE | Stereoacuity (arc seconds) | | Cover test (PD) Distance/Near | | | NFV (PD) Distance Near | | Prism Rx (PD) | | Duration (Months)/ mode of therapy |
|---|---|---|---|---|---|---|---|---|---|---|---|---|
| | | | Pre | Post | Pre | Post (with prism) | Post (without prism) | Pre | Post | Pre | Post | |
| 1 | 13/M | +1.00/ +1.00 | S | S | 12/10 | – | 8/0 | – | – | – | – | 3-6/HVT |
| 2 | 23/F | −4.50/ −4.00 | 70 | 40 | 16/12 | – | NR | 0 6 | 4 4 | – | – | <1/IVT |
| 3 | 10/M | −5.50/ −3.50 | 70 | 20 | 16/16 | – | 14/ 16 | 1 1 | 18 16 | – | – | <1 & 1–3/IVT & HVT |
| 4 | 36/M | 0.00/ 0.00 | 20 | 20 | 18/20 | – | 0/0 | 4 6 | 12 14 | – | – | <1/IVT |
| 5 | 23/M | −3.25/ −2.75 | 50 | 30 | 20/20 | – | 18/18 | 0 0 | 0 4 | – | – | <1/IVT |
| 6 | 32/M | −1.50/ −1.25 | 50 | 50 | 20/8 | 0/0 | 8/8 | 0 – | 6 16 | 8 | 8 | 1-3/HVT |
| 7 | 37/M | −1.50/ −1.50 | 40 | 20 | 25/20 | 8/6 | 8/6 | 0 0 | 7 – | 16 | 0 | <1/IVT |
| 8 | 24/F | −2.00/ 0.00 | 40 | 40 | 25/16 | 20/12 | 20/12 | – | – | 10 | 0 | 3-6/HVT |
| 9 | 15/M | −7.75/ −7.75 | S | 30 | 35/40 | 12/ 20 | 22/10 | 0 0 | 4 – | 20 | 10 | <1& 1–3/IVT& HVT |
| 10 | 25/M | −4.00/ −4.00 | S | NR | 40/40 | NR | NR | 0 0 | – | 24 | 12 | <1/IVT |
| 11 | 28/F | −4.50/ −5.50 | S | NR | 40/40 | 2/3 | 32/33 | 0 0 | 6 12 | 30 | 30 | 1-3 & 3–6/IVT & HVT |
| 12 | 15/F | −5.50/ −5.75 | S | 30 | 45/45 | 30/30 | 40/40 | 0 0 | 1 8 | 20 | 10 | <1/IVT |
| 13 | 19/M | 0.00/ 0.00 | S | 100 | 45/45 | 10/6 | 20/16 | 0 0 | 7 – | 35 | 10 | >6/HVT |
| 14 | 11/F | +1.50/ +1.25 | S | >400 | 45/55 | 0/10 | 30/40 | 0 0 | 7 – | 30 | 30 | 1-3/HVT |

SE: spherical equivalent of the refractive error, DS: diopter sphere, RE: right eye, LE: left eye, S: Suppression, PD: prism diopters, M: male, F: female, HVT: home vision therapy, IVT: in-office vision therapy, NR: not recorded.

The average duration of digital device use was 7.1±3.9 hours in our cohort. Seven out of nine patients reported having the habit of using the smartphone in supine position, while the other two did not. This history was not documented in the remaining five patients. Nine patients underwent took in-office vision therapy, and five patients took home vision therapy.

Negative fusional range (divergence) was measured with the prism correction (if any) to ensure the patient did not experience double vision during the test. Fig 3 shows the negative fusional vergence range for distance in 11 patients. Data was not available for others. A statistically significant improvement in the divergence was found for both distance (Wilcoxon signed rank, Z=−2.2, p=0.027) and near (Wilcoxon signed rank, Z=−1.9, p=0.046) after vision therapy. Median improvement of divergence was 7PD (range: 0–17PD) for distance and 8PD (range: 0–14PD) for near. At the end of therapy, one patient (case 11, in Table 2) did not show any improvement in divergence or reduction in eso deviation. This patient opted for surgical management. The longest follow-up was for 1 year (case 13) and the patient maintained their improved binocularity with Brock string exercise and prism spectacles (Fig 4). Three other patients had follow-up visits ranging from 2–5 months and demonstrated a stable status as well.

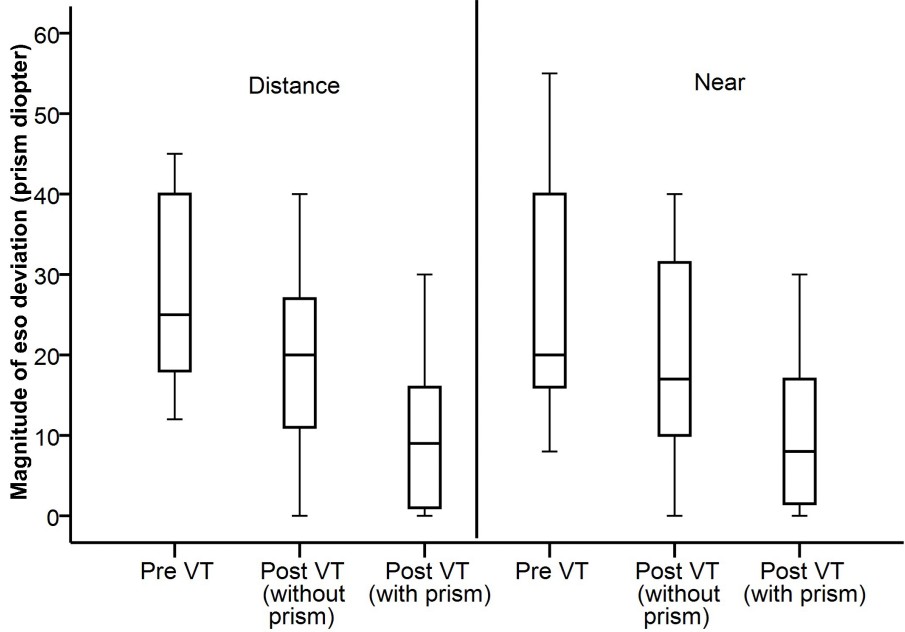

**Fig 2. Boxplot of the cover test values pre- and post-vision therapy (VT) for both distance (D) and near (N) for 12 patients.** Cover test values with and without prisms are also shown.

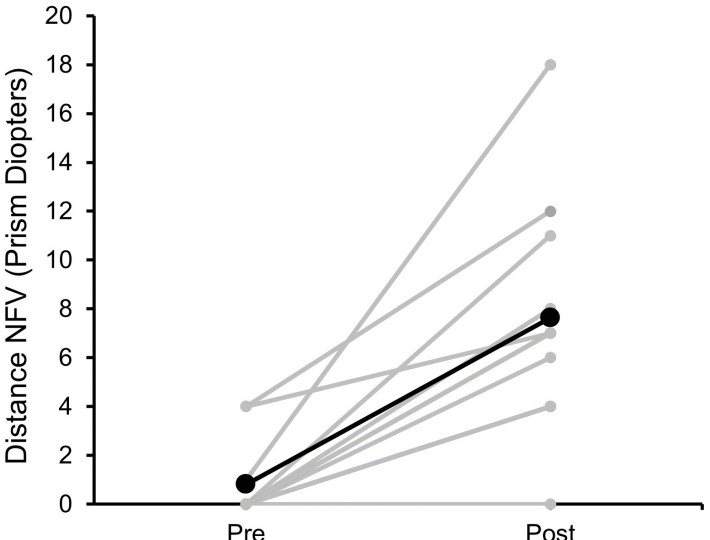

**Fig 3. Negative fusional vergence (NFV) break point for distance, measured pre- and post-vision therapy for each individual subjects (grey lines).** The thick black line represents the average NFV. Overall, an improvement in NFV values is seen.

## Discussion

This retrospective study investigated the outcomes from non-surgical management of AACE with prisms and/or vision therapy in 14 patients. Eso deviation and divergence range were the outcome measures after the management. We

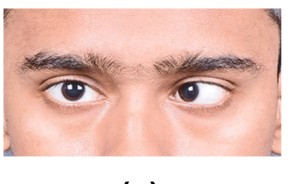 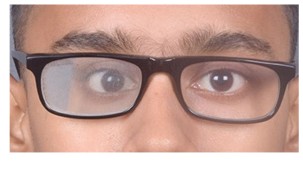 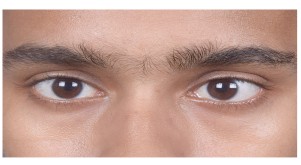

**(a)** **(b)** **(c)**

**Fig 4. Photograph of a patient (a) presenting with acute acquire esotropia, who was given (b) Fresnel prism spectacles.** Same patient (c) at the end of therapy, shows a reduction in the eso deviation.

observed that none of the patients experienced double vision with prisms and/or vision therapy. Eso deviation of 20PD or less was managed with vision therapy alone to achieve stable binocular single vision (n = 5). For those requiring prisms (64%, n = 9), single vision was achieved with prism power less than the eso deviation. The prism power was reduced (n = 4) or removed (n = 2) with vision therapy exercises in some of these patients (Table 2). The distance eso deviation was 25PD in two of the patients for whom the prism was eventually removed. Only in three patients (21%) the magnitude of the initial prism power was not reduced.

A reduction in eso deviation and an improvement in divergence range were observed in the present study (Figs 3 and 4). In both these parameters the median reduction for eso (6.5PD) and improvement for divergence (7PD) were comparable. While evidence for the effectiveness of vision therapy is well established for convergence insufficiency [18], it is not the case for divergence. It is believed that the first line of management for divergence insufficiency is to prescribe prisms [15]. The results of this study show that divergence can also be improved with vision therapy exercises. If the diplopia is constant, divergence therapy can be facilitated with the help of prisms, to initially achieve binocular single vision, after which divergence can be trained. The majority of the patients in this cohort (68%) had basic eso deviation, followed by divergence insufficiency (25%). Unlike exo deviation that is more bothersome for near work, with eso deviation, the distance double vision is more bothersome. This may prompt the patient to avoid distance tasks resulting in more near work, which could further vergence adapt them into the eso posture. Hence, relieving them of the double vision for distance, with advice to minimize near work or increase the near working distance (i.e., move objects further away) is a practical solution.

In a recent study [1] prisms were prescribed and then tapered in patients with AACE, for eso deviation of 25PD or lower. In our study we observed that prisms were mostly not needed if the deviation was 20PD or less. Improvement was obtained with only vision therapy exercises. Additionally, it was found that larger deviations (>25PD) can also be managed with a combination of both prisms and vision therapy exercises. However, a residual eso deviation remained in these patients. In recalcitrant cases where the eso deviation is not reduced, surgical management can be considered.

The Fresnel prism was split between the two eyes, to make it both economical and easier to taper the prism. Bilateral Fresnel prism was considered in an earlier study [10]. However, that study did not describe prism tapering. In general, the blur induced by the Fresnel prism may not be tolerated by patients. However, given the temporary nature of these prisms, and the fact that it was expected to be reduced, our patients did not mind the blur. For those patients who required the residual prism power, the final Fresnel prism power was prescribed to only one eye. This way the other eye (usually dominant eye) can be kept blur free, promoting tolerance to the Fresnel prism spectacles. Ground prisms can also be considered for better acceptance (both for cosmesis and clarity). One patient (case 6, Table 2) in this study preferred ground prism spectacles.

The skewed gender ratio observed in this study has not been highlighted before. In a recent study on AACE [19], 65% of the patients were males, a percentage comparable to our study (68% males). The reason for this gender predisposition

is unclear. It has been observed that adult females tend to practice the 20-20-20 rule more than males [20]. However, the history on practice of the 20-20-20 rule was not documented in this study. Like earlier studies [1,10,19], the majority (71%) of patients had myopic refractive error (Table 2) and could perhaps be having a Bielschowsky type of AACE (Type II in the newer classification) [2,13].

Even though several precipitating etiologies have been described [13], the exact pathogenesis of AACE is unclear. Vergence adaptation is said to occur when a prolonged vergence posture is maintained. This could be possible either by wearing a prism for a prolonged time or by viewing at a near distance for a prolonged time [21,22]. The dynamics of vergence eye movements also changes when vergence adaptation occurs [23,24]. Hence, prolonged near work could result in vergence adaptation, and this could be abnormal in individuals with binocular vision anomalies [25]. Such an adaptation could reset the baseline vergence status of an individual. Such a mechanism could be considered similar to that of accommodative spasm that could be triggered with close work [26]. Spasm of near reflex (SNR) involves both accommodation and vergence of the near triad. SNR with only the accommodative component has been found to be more common than its vergence counterpart [12]. It is possible that the vergence component presents as AACE. The underlying mechanism of SNR spectrum could have a shared etiopathogenesis, that could also involve AACE. This speculation needs further careful examination.

The average duration of the smartphone or digital device use in this cohort seems to be comparable to the general population [20]. The viewing distance for the digital device use was not documented in these patients. It is known that a closer viewing distance could increase the asthenopic symptoms [27]. While the question of smart phone use in supine position was asked, it is not known how many in the general population adapt such a posture while using their phones. Hence a concrete argument cannot be made that the supine position triggers AACE. A small but real effect was found in ocular parameters in different head postures [28]. While these experimental studies were done for relatively shorter duration, in real world scenario, the digital device use may last hours. Hence the effect could be even greater with the complex interplay through the cross-links of AC/A and CA/C. All this needs further investigation.

The retrospective nature of this study does pose some limitations. The history could have been more curated, perhaps with a checklist. There were some missing data, however, we do not think the overall trend would have been affected. Given that the patients came for different therapy durations or with different modality of therapies (home vs office-based vision therapy), some heterogeneity and variability is to be expected. Nevertheless, the overall trends of reduced eso deviation through objective measures, and improvements in divergence were both statistically significant and clinically meaningful. There was no control group included in this retrospective study. Ideally, this would have been patients who did not take prisms and/or vision therapy but observed for the natural course of the condition. However, given the duration of onset in the included patients ranging from few months to more than 1 year (Table 1), it is evident that the condition can be long standing and yet be amenable to the management protocol described here. This is the first study to show an effective management in some types of AACE using vision therapy exercises with or without prisms. These preliminary findings can help plan a larger prospective study.

## Conclusion

In conclusion, non-accommodative, non-neurologic types of AACE can be managed non-surgically through a combination of prisms and/or vision therapy in 79% of patients. Specifically, divergence ranges can be improved and eso deviation can be reduced with this management technique. For most patients with less than 20 prism diopter intermittent eso deviation, vision therapy alone will suffice. In few patients with eso deviation up to 25 prism diopter, the prisms could be removed after vision therapy. For higher magnitudes of deviation along with constant eso deviaiton, prisms can be prescribed to achieve single vision and later can be tapered with vision therapy exercises. The options of prisms and/or vision therapy can be considered especially for those patients who show variable angles of deviation or who are unwilling or unfit for surgery.

## Supporting information

**S1 File. Vision Therapy Protocol.**
(DOCX)

## Acknowledgments

Dr.Miriam Conway for reviewing this manuscript and giving critical feedback. Ms.Monika Thakur for formatting help.

## Author contributions

**Conceptualization:** PremNandhini Satgunam, Rohan Nalawade.

**Data curation:** PremNandhini Satgunam, Rinki Gupta, Manali Sikder, Rohan Nalawade, Ramesh Kekunnaya.

**Formal analysis:** PremNandhini Satgunam, Rinki Gupta, Manali Sikder.

**Investigation:** PremNandhini Satgunam, Manali Sikder, Rohan Nalawade, Ramesh Kekunnaya.

**Methodology:** PremNandhini Satgunam, Rinki Gupta, Manali Sikder, Rohan Nalawade.

**Project administration:** PremNandhini Satgunam, Ramesh Kekunnaya.

**Resources:** PremNandhini Satgunam.

**Supervision:** PremNandhini Satgunam, Ramesh Kekunnaya.

**Validation:** Rinki Gupta.

**Visualization:** PremNandhini Satgunam.

**Writing – original draft:** PremNandhini Satgunam, Rinki Gupta, Manali Sikder.

**Writing – review & editing:** PremNandhini Satgunam, Rinki Gupta, Manali Sikder, Rohan Nalawade, Ramesh Kekunnaya.

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
