## [Decision Letter · Decision Letter 0]

25 Jun 2025

PONE-D-25-10143Management of Acute Acquired Comitant Esotropia Using Prisms and Vision TherapyPLOS ONE

Dear Dr. Satgunam,

Thank you for submitting your manuscript to PLOS ONE. After careful consideration, we feel that it has merit but does not fully meet PLOS ONE’s publication criteria as it currently stands. Therefore, we invite you to submit a revised version of the manuscript that addresses the points raised during the review process. Please submit your revised manuscript by 09 August 2025. If you will need more time than this to complete your revisions, please reply to this message or contact the journal office at plosone@plos.org . Please include the following items when submitting your revised manuscript:

We look forward to receiving your revised manuscript.

Kind regards,

Liat Gantz, PhD

Academic Editor

PLOS ONE

Journal Requirements:

“Hyderabad Eye Research Foundation for their support to carry out the project.”

Additional Editor Comments:

The manuscript could benefit from English editing, with some minor suggestions below.

Additional comments specified below include lack of references, lacking details in the methods section, and ambiguity in Figure 3

Introduction

Lines 2,3- require references

Line 5- however in recent years, its incidence has been on the rise

Line 16- in the interim (without time)

Line 17- remove “their eye”

Line 23- previously demonstrated binocular single vision, training with vision therapy to revert to the prior condition, renders vision therapy plausible.

Line 35- amongst these patients

Line 42- The methods described herein are the typical clinical tests conducted in this clinic.

Line 45- which drug? And how many drops per eye?

Line 49- to examine distance fusion

Line 50- provide a references that the red lens test has a lower dissociation compared to the Worth four dot test.

Line 51- the red lens test is typically performed with a penlight or spot of light. It is unclear why the patient would be expected to report several charts

Line 53: red tint?

Lines 55-57: provide a reference for this (relationship between phoria at distance and near to determine the type of impairment)

Line 83: provide a reference for Fresnel prism hampering contrast and reducing visual acuity

Line 88: why is dispensing a Fresnel sticker for both eyes more economical that just one eye? Doesn’t it cost double the amount?

Figure 1: below instead of down

Line 96: I believe it is an aperture rule and not ruler

Line 100: for a specific time at home? Or everyone does however much they want?

Line 100: those who did not come and were given Brock String 20 min per day twice a day- were they given a tutorial? How did they know what to do with the Brock string?

Line 123 – performed

Line 127: Table 1 displays, delete the word “all”

Line 130: arrived to their follow-up appointment

Line 131: did not participate in vision therapy

Line 135: five (spell out numbers lower than 10)

Line 138: preferred instead of wanted

Line 141: remove “with therapy” as you wrote “After vision therapy” in the beginning of the sentence

Line 149: two and five, spell out nine

Figure 3: Is this the break point? When you say “range”- do you mean the average of break and recovery? Not clear..

Line 158: three

Line 178: The majority

Line 229: was found

Line 243: ranging from

Reviewers' comments:

Reviewer's Responses to Questions

**Comments to the Author**

1. Is the manuscript technically sound, and do the data support the conclusions?

Reviewer #1: Yes

Reviewer #2: Yes

Reviewer #3: Partly

2. Has the statistical analysis been performed appropriately and rigorously? 

Reviewer #1: Yes

Reviewer #2: Yes

Reviewer #3: Yes

3. Have the authors made all data underlying the findings in their manuscript fully available?

Reviewer #1: Yes

Reviewer #2: Yes

Reviewer #3: Yes

4. Is the manuscript presented in an intelligible fashion and written in standard English?

Reviewer #1: Yes

Reviewer #2: Yes

Reviewer #3: Yes

5. Review Comments to the Author

Reviewer #1: This is a well-written and insightful article. It highlights the importance of promoting non-surgical management approaches in similar cases. Articles like this not only enhance clinical understanding but also encourage practitioners to explore and implement conservative, evidence-based treatment options where appropriate.

Reviewer #2: This manuscript presents a valuable retrospective study examining non-surgical management of acute acquired comitant esotropia (AACE) using prisms and vision therapy. The authors report promising results, with 79% of patients showing improvement through this approach.

The paper is well-structured with a clear rationale, detailed methodology, and meaningful results.

Overall, this study provides clinically relevant evidence for an alternative management strategy for AACE and represents a worthwhile contribution to the field. I suggest some minor revisions.

- Sample Size and Study Design: Address the limitations of the small sample size (n=14) and retrospective design more critically. Consider discussing whether the findings should be considered preliminary until validated in a larger prospective study.

- Patient Selection Guidelines: a clearer clinical guidelines regarding which AACE patients are suitable candidates for non-surgical management versus those who should proceed directly to surgery would significantly enhance the practical application of your findings.

- Long-term Outcomes: if possibile provide data on the longest follow-up period available and discuss whether the improvements observed were maintained over time.

Reviewer #3: 1. Only patients with esotropia deviation and divergence range outcome measures pre and post management should be included in this manuscript (patients 1,2,8 &10 should be excluded). Hence, table 1, table 2, Fig 2 and Fig 3 should be all, accordingly, reedited.

2. It is advisable to add NFV (PD) at near data, if available. It might contribute for understanding the mechanisms of AACE and treatment success.

3. The classification mentioned in line 35 is based on research in pediatric population (reference 13). The population of the current manuscript includes adults. The authors should clarify it.

4. Line 53, was it simultaneous prism cover test or alternate cover test?

5. The VTS 4 is described, however, it is not clear which measurements with the VTS 4 and at what specific distance were included (for example, measuring subjective eye deviation at the mentioned testing distance of 84 cm is probably irrelevant). Furthermore, the gold standard for measuring NFV at distance and near is with prism bar (was it done?). Was the VTS4 validated for measuring NFV or deviation in children\adults?

6. The subgroup of patients who were able to complete the vision therapy (table 1 first line) were all in the age of pre-presbyopia and had a shorter duration of onset of the condition range. It is worth a discussion.

7. The authors should clarify the way they measured the post cover test with prisms in glasses. Using two horizontal stacked prisms can induce a measurement error (Thompson JT, Guyton DL. Ophthalmic prisms. Measurement errors and how to minimize them. Ophthalmology. 1983 Mar;90(3):204-10. )

8. It is advisable to add more detailed vision therapy protocol as an e-supplement .

6. PLOS authors have the option to publish the peer review history of their article (what does this mean? ). If published, this will include your full peer review and any attached files.

**Do you want your identity to be public for this peer review?** For information about this choice, including consent withdrawal, please see our Privacy Policy .

Reviewer #1: **Yes:** Preeti Sharma

Reviewer #2: **Yes:** prof. Aldo Vagge

Reviewer #3: No

---

## [Author Response · Author response to Decision Letter 1]

28 Jul 2025

We have uploaded a Response to Reviewers document

---

## [Decision Letter · Decision Letter 1]

1 Oct 2025

PONE-D-25-10143R1Management of Acute Acquired Comitant Esotropia Using Prisms and Vision TherapyPLOS ONE

Dear Dr. Satgunam,

Thank you for submitting your manuscript to PLOS ONE. It is clear that you put work into your revision and one reviewer felt that all comments were addressed. However, another reviewer felt that there were several issues that should still be revised. I have also added three questions to include in your revision below (editor comments). We invite you to submit a revised version of the manuscript that addresses the points raised during the review process.

We look forward to receiving your revised manuscript.

Kind regards,

Liat Gantz, PhD

Academic Editor

PLOS ONE

Journal Requirements:

Additional Editor Comments :

I commend the authors for their revision. One reviewer is satisfied and the other still requires some changes therefore I recommend a "minor revision". In addition to that reviewer's comments I have three items that I would like the authors to address:

1 The reviewer asked : The authors should clarify the way they measured the post cover test with prisms in glasses. Using two horizontal stacked prisms can induce a measurement error (Thompson

JT, Guyton DL. Ophthalmic prisms. Measurement errors and how to minimize them. Ophthalmology. 1983 Mar;90(3):204-10. )

The authors responded: Thank you for this question. Yes, stacking up prisms especially when held in the same direction gives error mainly due to the air gap between prisms. In our patient cohort, only Fresnel prism sheet was used. At the end of therapy, patients had the Fresnel prism worn over one eye only. So neutralizing prisms (plastic prism) can be held over the other eye. We have also observed that holding the prism in front of the eye with Fresnel prism sheet, also gives comparable results.". I would add this to the discussion for other readers who may have similar questions as they read the manuscript

2 the authors state that cycloplegia was applied prior to the binocular vision examination, which included stereopsis. Is stereopsis, which is measured at 40 cm and requires 2.5 D of accommodation (approximately) expected to be within normal limits if the patient is cyclopleged? I did not see any stereopsis findings in the tables provided? Also, would any stereopsis be expected in patients with strabismus?

3 The authors stated that "If the eso deviation for distance and near were comparable within 4PD (prism diopter), it was termed basic eso deviation." - surely there is a minimal amount that is considered "normal"? If one has 2^ eso at distance and near, is it "basic eso"? Similarly, If the eso deviation was greater for distance than near, it was termed divergence insufficiency, vice-versa was convergence excess. [15]- also here specify the minimal amounts.

Reviewers' comments:

Reviewer's Responses to Questions

**Comments to the Author**

1. If the authors have adequately addressed your comments raised in a previous round of review and you feel that this manuscript is now acceptable for publication, you may indicate that here to bypass the “Comments to the Author” section, enter your conflict of interest statement in the “Confidential to Editor” section, and submit your "Accept" recommendation.

Reviewer #3: All comments have been addressed

Reviewer #4: (No Response)

2. Is the manuscript technically sound, and do the data support the conclusions?

Reviewer #3: Yes

Reviewer #4: Partly

3. Has the statistical analysis been performed appropriately and rigorously? 

Reviewer #3: Yes

Reviewer #4: N/A

4. Have the authors made all data underlying the findings in their manuscript fully available?

Reviewer #3: Yes

Reviewer #4: No

5. Is the manuscript presented in an intelligible fashion and written in standard English?

Reviewer #3: Yes

Reviewer #4: Yes

6. Review Comments to the Author

Reviewer #3: (No Response)

Reviewer #4: In this study, the results of prism and/or vision therapy in 14 patients with AACE were reviewed and it was found that the divergence range can be improved and the esodeviation can be decreased with this management technique. In 7 of the 14 cases, prism treatment could be discontinued after prism and/or vision therapy. However, there are also problems:

1、 Of the 14 patients, 5 had variable deviation or intermittent diplopia. Should myasthenia gravis and accommodative spasm be excluded?

2、 Two classification methods of AACE were mentioned in the study. One was used for inclusion and the other for discussion (line 217). Why is the same literature used for citation? In addition, among the 14 patients, none were under 10 years of age and 11 patients were over 14 years of age. However, the use of the AACE classification for children as the inclusion criteria of the study is still questionable.

3、 Table 2 should be supplemented with the classification of AACE, the accommodation measures, the prism diopters worn and the prognosis.

4、 Of the 9 cases treated with prisms, Fresnel prisms were prescribed in 8 cases and group prisms in 1 case. However, only the Fresnel prism was mentioned in the ABSTRACT.

5、 In the 9 patients with a deviation of more than 20PD, prism treatment was first prescribed to eliminate the diplopia. The prisms were evenly distributed to both eyes. What was the prescribed amount of prisms? After the deviation was reduced by vision therapy, the prisms in the non-dominant eye were removed, i.e. the prisms were reduced by half. Were all six patients able to eliminate the diplopia? How is it possible to gradually reduce the prism?

6、 Are there any results after stopping the vision therapy for three or six months?

7、 The treatment results of the five patients who did not wear prisms were not mentioned in the RESULTS section.

8、 In the RESULTS section, it was mentioned that some patients were lost to follow-up. What were the reasons for the loss to follow-up? Is it because the treatment effect is not good?

9、 In the DISCUSSION section, it was mentioned that SNR might be a pathogenic factor of AACE. Are there any results of accommodation measures?

10、 The conclusion mentioned that 79% (11 cases) of patients had a reduction in prism degree after vision therapy. This should include the 5 cases that did not wear prisms, but the deviation of cases 3 and 5 did not decrease significantly.

11、 All 5 cases that did not need to wear prisms achieved the disappearance of diplopia after vision therapy. However, the deviations in 2 cases did not decrease significantly, and the fusion amplitude of case 5 did not increase either. How can it be explained that these two patients were able to eliminate diplopia?

12、 It is recommended to provide photos of ocular alignment before and after treatment for some patients, as this is more intuitive.

7. PLOS authors have the option to publish the peer review history of their article (what does this mean? ). If published, this will include your full peer review and any attached files.

**Do you want your identity to be public for this peer review?** For information about this choice, including consent withdrawal, please see our Privacy Policy .

Reviewer #3: No

Reviewer #4: No

---

## [Author Response · Author response to Decision Letter 2]

25 Oct 2025

Response to reviewers is attached as a word document

---

## [Decision Letter · Decision Letter 2]

25 Nov 2025

PONE-D-25-10143R2Management of Acute Acquired Comitant Esotropia Using Prisms and Vision TherapyPLOS ONE

Dear Dr.  Satgunam, Thank you for submitting your manuscript to PLOS ONE. After careful consideration, we feel that it has merit but does not fully meet PLOS ONE’s publication criteria as it currently stands. Therefore, we invite you to submit a revised version of the manuscript that addresses the points raised during the review process.

We look forward to receiving your revised manuscript.

Kind regards,

Liat Gantz, PhD

Academic Editor

PLOS ONE

Journal Requirements:

Additional Editor Comments:

We are sorry for the delay, but securing reviewers to re-review took some time.One reviewer feels that the conclusions should be modified. I think the paper should be read from beginning to end to ensure logical flow. Professional English language editing or reviewing by an English speaking colleague would greatly benefit the final version.  We invite you to submit a revised version of the manuscript that addresses the points raised during the review process. Please ensure that your decision is justified on PLOS ONE’s publication criteria  and not, for example, on novelty or perceived impact.

Thank you to the authors for revising the submission. I am sorry for the delay- it was difficult to secure re-review for this re-submission.

Note that one reviewer that required revisions feels that the conclusion remains insufficiently objective and lacks accuracy. The details are as follows:

1. The conclusion should emphasize that prisms and/or vision therapy are intended for patients with variable deviations or for those who are unwilling or not ready to undergo surgery.

2. Prisms and/or vision therapy are helpful for 79% of patients with variable deviations or those not ready for surgery. However, not all of these patients experienced a reduction in deviation. The true clinical value lies in the disappearance of diplopia or its resolution after discontinuing the prism following prisms and/or vision therapy, which occurs in 50% of cases.

3. Among the 9 patients with constant deviations greater than 20 prism diopters, 2 cases (22%) achieved binocular single vision without prisms after treatment. Both of these cases had deviations of 25 prism diopters. This should also be noted in the conclusion.

I would like to add that the authors should read the manuscript from beginning to end to make sure it flows logically.

For example: I asked the authors how the stereopsis was measured with cycloplegia. They responded that the binocular visual examination was carried out on a separate visit and this was supposedly addressed in the revision. However-

read this section (Lines 47-51):

"All patients underwent cycloplegic refraction (one drop of 1% cyclopentolate hydrochloride and one drop of tropicamide plus (0.8% tropicamide +5% phenylephrine hydrochloride)), prior to the binocular vision assessment. "

Note that it says there was cycloplegia PRIOR (i.e. before) the binocular vision assessment.

Then the authors write in the subsequent sentence " Binocular vision assessment was performed on a separate visit, with the best refractive correction."

This seems to contradict the prior sentence.

Then the subsequent sentence says "The measurements include"- do the authors mean that the binocular vision examination procedures included: (in past tense)?

The grammar must be fixed here:

Line 52: fusion with the Worth four dot test at 40 cm and a red les test over one eye to examine distance fusion. Add "A" before "red lens"

Line 56: what does "the color for the same" mean?

Line 56: "under normal viewing conditions" instead of "in normal viewing condition"

Line 60: it was termed a basic eso deviation. Must add "a"

Line 61: delete the "vice-versa it was convergence excess" and write out the sentence. It makes no sense in its current form.

Line 71: the Muscle Imbalance - add "the"

Line 72: can be obtained or measured instead of "achieved"

Line 73: a prism bar- add "a"

Line 78: instead of "it will not" which is unclear please specify if the authors mean to say that on one side the lines do not appear as a plus?

line 78- watch the tenses here. "is detected" should be followed by "is increased" and not "was increased" since later the patient "reported " (past)- perhaps just change everything to past tense? If the odd one WAS detected correctly, the vergence demand was increased automatically until the patient reported seeing two dolphins, at which point the judgment of the odd one was difficult. (delete "also")

Line 81: until instead of "till"

Line 91: since you modified the protocol (in the past)- the Fresnel sheet was cut into two

Line 93: one 15PD Fresnel sheet was cut and dispensed such that each eye received 15PD Base out.

Line 94: and easy to dispense

Line 95: when the patient improved

Line 96: vision therapy, it was possible to peel off one of the Fresnel sheets to reduce the prismatic correction by half.

Line 97-98: After the prismatic correction was reduced, the patient was dispensed divergence exercises.

Line 98- During subsequent visits, complete removal or reduction of prism magnitude were considered.

Figure 1 caption: The VTS4 system includes a stere-television with dichoptic targets viewed through stereo-goggles. (a) phoria measurement requiring alignment of the blue cross in the center of the red circle using an XBox. The inset shows the magnified view of the dichoptic targets. (b) ....the patient was asked to identify the odd side. In this image, the odd target is below. In the remaining three slides, a red plus is seen with fusion.

Line 102: included

Line 103: delete the word "given" - just exercises at different viewing distances

Line 105: systems (plural)

Line 108: the Brock string exercise

Line 113: until reaching the diplopia threshold

Line 114: they were encouraged to attempt to fuse the bead into a single bead

Lines 118-120: Vergence rock exercises were also combined with the Brock String. Patients were asked to look at the three beads at different distances, while maintaining clear and single vision of each bead for approximately 20s, while experiencing physiological diplopia.

Lines 129-130: All measurements were assessed at distance (3m) and near (40cm).

Line 130: A non-parametric test for paired comparisons...

Line 135: during the study period of 10 months.

Line 140-141: All of these patients

Line 147: non-constant

Line 149: was instead of were

Line 154: Both of these reductions

Line 162: underwent instead of "took"

Line 163: negative fusional range (divergence) was measured with prism correction (if any) to ensure that the patient did not experience diplopia during the testing

Line 185: was not reduced

Line 186: Not clear which study is "this study"

How about: Additionally, reductions in eso deviation and improvement in divergence were observed.

Line 187: What is the meaning of "Synchronously"- replace

Line 209: However, that study did not describe prism weaning.

Line 212: it was expected to be reduced

Line 214: prescribed to only one eye.

Line 214: This way, the fellow eye (usually the dominant eye) was kept blur-free, promoting tolerance to the Fresnel prism spectacles.

Line 216: Ground prisms. Delete "else"'

Line 217: preferred ground prism spectacles.

Line 246: Previous studies describe short duration experiments, whereas in real-world conditions, digital device may last hours.

Line 249 AC/A and CA/C, requiring further investigation.

Line 256-257: A control group was not included in this retrospective investigation.

Line 267: ranges

Reviewer's Responses to Questions

**Comments to the Author**

1. If the authors have adequately addressed your comments raised in a previous round of review and you feel that this manuscript is now acceptable for publication, you may indicate that here to bypass the “Comments to the Author” section, enter your conflict of interest statement in the “Confidential to Editor” section, and submit your "Accept" recommendation.

Reviewer #4: (No Response)

2. Is the manuscript technically sound, and do the data support the conclusions?

Reviewer #4: Partly

3. Has the statistical analysis been performed appropriately and rigorously? 

Reviewer #4: N/A

4. Have the authors made all data underlying the findings in their manuscript fully available?

Reviewer #4: Yes

5. Is the manuscript presented in an intelligible fashion and written in standard English?

Reviewer #4: Yes

6. Review Comments to the Author

Reviewer #4: (No Response)

7. PLOS authors have the option to publish the peer review history of their article (what does this mean? ). If published, this will include your full peer review and any attached files.

**Do you want your identity to be public for this peer review?** For information about this choice, including consent withdrawal, please see our Privacy Policy .

Reviewer #4: **Yes:** Jingchang Chen

---

## [Author Response · Author response to Decision Letter 3]

4 Dec 2025

Response to Reviewer document is attached

---

## [Editor Report · Decision Letter 3]

15 Dec 2025

Management of Acute Acquired Comitant Esotropia Using Prisms and Vision Therapy

PONE-D-25-10143R3

Dear Dr. Satgunam,

We’re pleased to inform you that your manuscript has been judged scientifically suitable for publication and will be formally accepted for publication once it meets all outstanding technical requirements.

Kind regards,

Liat Gantz, PhD

Academic Editor

PLOS One

Additional Editor Comments (optional):

The authors addressed the minor comments and the paper is now ready for publication.
---

## [Editor Report · Acceptance letter]

PONE-D-25-10143R3

PLOS One

Dear Dr. Satgunam,

I'm pleased to inform you that your manuscript has been deemed suitable for publication in PLOS One. Congratulations! Your manuscript is now being handed over to our production team.

Kind regards,

on behalf of

Dr. Liat Gantz

Academic Editor

PLOS One